precision diagnostics; precision medicine; omics; machine learning; children

**Author for correspondence:**
Paul Dimitri,
Email: paul.dimitri@nhs.net

# Precision diagnostics in children

## Paul Dimitri[1,2]

[1]Department of Paediatric Endocrinology, Sheffield Children's NHS Foundation Trust, Sheffield, UK and [2]The College of Health, Wellbeing and Life Sciences, Sheffield Hallam University, Sheffield, UK

## Abstract

Medical practice is transforming from a reactive to a pro-active and preventive discipline that is underpinned by precision medicine. The advances in technologies in such fields as genomics, proteomics, metabolomics, transcriptomics and artificial intelligence have resulted in a paradigm shift in our understanding of specific diseases in childhood, greatly enhanced by our ability to combine data from changes within cells to the impact of environmental and population changes. Diseases in children have been reclassified as we understand more about their genomic origin and their evolution. Genomic discoveries, additional 'omics' data and advances such as optical genome mapping have driven rapid improvements in the precision and speed of diagnoses of diseases in children and are now being incorporated into newborn screening, have improved targeted therapies in childhood and have supported the development of predictive biomarkers to assess therapeutic impact and determine prognosis in congenital and acquired diseases of childhood. New medical device technologies are facilitating data capture at a population level to support higher diagnostic accuracy and tailored therapies in children according to predicted population outcome, and digital ecosystems now tailor therapies and provide support for their specific needs. By capturing biological and environmental data as early as possible in childhood, we can understand factors that predict disease or maintain health and track changes across a more extensive longitudinal path. Data from multiple health and external sources over long-time periods starting from birth or even in the *in utero* environment will provide further clarity about how to sustain health and prevent or predict disease. In this respect, we will not only use data to diagnose disease, but precision diagnostics will aid the 'diagnosis of good health'. The principle of 'start early and change more' will thus underpin the value of applying a personalised medicine approach early in life.

## Impact statement

There are 1.8 billion young people in the world today – 40% of the global population is under 24. We should aim to support children and young people with maintaining a long and healthy life and develop a system by which we personalise their health and healthcare for a better future. New advances in the fields of genomics, proteomics and cytogenetics have revolutionised our ability to understand the mechanisms underpinning the evolution and manifestation of existing diseases, discover new diseases, develop targeted therapies for specific populations of children to improve outcomes and predict children's response to therapy, their risk of relapse and their prognosis. Importantly, our ability to diagnose disease early in life, and in particular using genomics in newborn screening supports early intervention and prevention and provides a greater understanding of the evolution and manifestation of diseases over longer periods. Critical to this process is the study of disease in the context of exposome, a new paradigm that encompasses the totality of human environmental exposures from conception onwards. A personalised approach to children's health requires knowledge of the patient as an individual and their surrounding ecosystem. New technologies that can provide accurate tracking and recording of environmental data and technologies that can generate predictive models relating to future outcomes will add to the factors that facilitate a precision-based approach to managing health and disease over the life course. Furthermore, the ability to capture health data across large populations of children allows us to understand population characteristics in specific diseases that in turn drive individual interventions and changes. Precision medicine, precision diagnostics and the technological advances underpinning these rapidly advancing fields will change the way in which we understand disease, sustain health and improve quality of life. Focussing our efforts early in life is an investment for future health and healthcare.

## Introduction

Healthcare is evolving from a traditional 'one-size-fits-all' approach to a model that accounts for the prediction of individual patient disease risks, a tailored approach to the investigation and the development of targeted interventions. A better understanding of the underlying mechanisms

and causes of rare and chronic diseases combined with technological advancements are transforming medicine from a reactive to a pro-active and preventive discipline that has been encapsulated in the '4P' approach to medicine (also referred to as P4 Medicine) to maintain health and well-being and to prevent and predict disease and thus respond accordingly to individual needs – '4P – Preventive, Predictive, Participatory and Personalised' (Hood et al., 2004; Flores et al., 2013). A personalised approach to medicine requires knowledge of the patient as an individual and their surrounding ecosystem, by understanding their molecular and genetic components, their cells and tissues, the whole person and the population and environment in which they exist. This breadth and depth of knowledge of an individual and their environmental interaction underpins the science of systems biology (National Research Council (US), 2009). Systems biology is founded on the principle that the interaction between individual parameters at the genetic, molecular and cellular level with social and environmental parameters as the component parts of complex biological systems supports a more 'predictive' approach by which computational models can be used to predict outcomes, and individuals can be stratified more comprehensively according to their disease risks. Systems medicine focuses on the approach of systems biology to human disease by combining knowledge that ranges from individual genomic sequencing to understanding global data sets that track patient populations and their interaction with the environment (Hood et al., 2012).

Advances in technology and computational analysis over the last 20 years have radically enhanced our ability to collect, store and analyse data to create biological networks derived from computational models that demonstrate how factors within biological systems can maintain health or lead to disease. Importantly, factors within these computational models can be modified to predict an individual's risk of future ill-health. Ultimately, the aim of systems medicine is to develop an 'individual data profile' founded upon multi-dimensional longitudinal health data, ranging from the individuals genomic/proteomic profile, hospital data from historic interactions and lifestyle and environmental data. These data have two fundamental origins – data that are derived when an individual interacts with a system such as a hospital or for health research, or alternatively, when the individual offers or inputs the data as part of an interaction with a digital platform encompassed in the participatory component of the 4P model. The latter of these has been facilitated by the advance in technologies that allow individual consumers to input and track their own health data and interact with others to monitor their health and disease risk. Portable devices such as mobile phones, tablets and laptops, novel sensors and digital platforms help to facilitate the participatory component of the 4P model extending health data collection and analysis into homes, workplaces and schools. Data collection from new devices can be passive, such that the individual allows the device to track activity without direct input, or active by which the individual inputs the data. In principle, if information can be collected early in life, this has the potential to provide richer longitudinal data about health, disease, lifestyle and environmental interaction, which could help modify factors that prevent disease, support earlier diagnoses or maintain health. Returning to the systems medicine approach, data could be derived from a number of sources including primary and secondary care health data, genomics data derived from genomic newborn screening and whole genome sequencing (WGS), educational data from schools and universities, and additional data on lifestyle and environmental exposure, sometimes referred to as the exposome, encompassing environmental

exposure from the prenatal period onwards. For children, the exposome encompasses general exposure to the external environment such as climate, familial and social factors and education, and specific exposure such as infections, radiation, home factors such as diet, smoking and alcohol consumption in parents, and physical activity. The challenge as we develop tools to individualise healthcare and to prevent and predict disease in children and young people is our ability to acquire data from multiple sources including those which are controlled by the consumer. Data captured from an early age will rely on empowering children, young people and their families to permit the use of their data, particularly data that may be from non-health related and personalsources, trusting that their data will be utilised appropriately to support their health and well-being and will be stored securely (National Research Council (US), 2011). The value of using data from multiple sources must be easily demonstrable to families in a way that prevention, disease modification and individualised care in the future are supported by meaningful and actionable information that is in a way that children, young people and parents can understand, and in a way that parents can utilise to support their children in preventing ill-health or seeking earlier healthcare support.

Precision diagnostics is a branch of precision medicine using the 4P approach by which individual diseases are diagnosed based upon genomic variation, lifestyle and the environment. Understanding subpopulations with similar genomic information and the environmental and lifestyle factors that predict disease onset allows healthcare professionals and individuals to prevent disease onset or intervene early to prevent disease progression. This approach has the potential to develop a radically more cost-effective approach to healthcare, particularly in the paediatric population, where rare diseases are more prevalent. Moreover, a better understanding of modifiable lifestyle factors in childhood may help to predict or prevent future disease, thus underpinning the value of prospective data collection. Tailored therapies by stratifying individuals into subgroups according to their disease profile, response to therapy and prognosis and preventing disease through risk factor modification will result in a reduction in healthcare expenditure (Wang et al., 2017). Furthermore, understanding the molecular and cellular origins of disease in children and young people has resulted in a paradigm shift in thinking about 'causes' rather than 'symptoms' of disease leading to a proposal for a new taxonomy for disease classification that could change the approach to clinical decision-making, diagnosis, therapy and prognosis (Blower et al., 2020).

## Disease reclassification and predicting future disease in children

Precision diagnostics is founded on the ability to predict disease and the ability to adopt a personalised approach. The understanding of a disease at a molecular level will determine the onset of the disease or symptoms, or when a perturbation within a biological system will result in the manifestation of the disease. This will vary between individuals, and thus, the understanding of the biological environment at a molecular level is fundamental in determining the individual's disease susceptibility, the severity of the disease, response to specific therapies and prognosis. At a genetic level, mutations range from loss or gain of entire chromosomes, loss of smaller regions of DNA, for example, copy number variants (CNVs), to changes in the structure of the genome in the form of translocations, inversions and insertions, or changes in the sequence of the nucleotides

(Lalonde et al., 2020). Advances in genomics, and in particular the development of next-generation sequencing (Lalonde et al., 2020), have had a transformative effect on diagnostics in children, resulting in targeted panels for specific diagnoses, and examination of the entire exome or genome based upon clinical manifestations. Whole exome sequencing and Whole Genome Sequencing (WGS) have the potential to increase the diagnostic yield by up to 50% and as such there has been a call to adopt WGS as a first-tier test, particularly conditions in children with genetic heterogeneity (Stavropoulos et al., 2016; Meng et al., 2017; Posey et al., 2017; Lionel et al., 2018). Many of these conditions are congenital, manifesting at birth, infancy or early childhood. A genetic diagnosis can then inform prognosis, anticipatory care and surveillance, targeted management and future family planning. Even within conditions harbouring the same 'macro' genetic mutation, there can be a high degree of variability based upon the location of the specific mutation and the interaction with other genes within the same molecular pathway (Costain et al., 2020). In turn, this allows individualised planning, targeted therapies and risk profiling.

This approach has led to significant clarification in disorders with heterogeneous manifestations which present in childhood, and the reclassification of disease based upon genotype rather than clinical presentation. For example, the heterogeneous clinical manifestations of Noonan syndrome, a condition that presents with a distinct facial phenotype, short stature, cardiac anomalies, developmental delay and other features (Dahlgren and Noordam, 2022), have been reclassified and incorporated into a group of genetic mutations that are collectively known as the RAS-opathies (Gripp et al., 2020). RAS-opathies define germline mutations in genes that encode components or regulators of the Ras/mitogen-activated protein kinase pathway (Rauen, 2013) and share many of the phenotypic manifestations of Noonan syndrome. RAS-opathies also include neurofibromatosis type 1, Noonan syndrome with multiple lentigines, Noonan syndrome-like disorder with loose anagen hair, Noonan syndrome-like disorder with or without juvenile myelomonocytic leukaemia, capillary malformation-arteriovenous malformation syndrome, Costello syndrome, cardio-facio-cutaneous syndrome, SYNGAP1-related intellectual disability and Legius syndrome. Mutations associated with Noonan syndrome alone include *CBL, BRAF, KRAS, LZTR1, MAP2K1 (MEK1), NRAS, PTPN11, RAF1, RIT1, SOS1* and *SOS2* (Riller and Rieux-Laucat, 2021; Leoni et al., 2022). Thus, this provides an example of one of many genetic disorders in children that may have been mistakenly classified according to facial and systemic phenotype, or previously described as 'non-classical' or 'atypical' which can now be accurately genotyped to define the appropriate disorder, the predicted consequences, therapy and prognosis.

Similarly, osteogenesis imperfecta (OI, brittle bone disease), a rare disease in childhood, was originally classified by Sillence in 1979 into four subtypes (OI I–IV) based upon mode of inheritance, clinical manifestation and severity (Sillence et al., 1979). However, 20 different subtypes of this condition have emerged based upon a more refined understanding of the wide range of genes that encode relevant proteins. These proteins are involved in processes such as the synthesis of type I collagen and the differentiation, regulation and activity of bone-forming cells (osteoblasts) and bone-absorbing cells (osteoclasts). Importantly, this has led to an exponential rise in targeted therapies for bone fragility in OI based upon the genotype-derived causative mechanisms, thus supporting the value of genomic reclassification and precision diagnostics to develop novel targeted therapies for rare diseases in children (Zaripova and Khusainova, 2020). The reduction in the cost of genomic

sequencing is facilitating significant advances in the classification of diseases but additionally in predicting the onset of disease. In the UK, Genomics England has embarked on a discovery science programme aiming to sequence and analyse the whole genome in up to 200,000 babies for a set of actionable genetic conditions which may affect their health in early years. This aims to ensure timely diagnosis, access to treatment pathways and enable better outcomes and quality of life for babies and their families (Genomics England, 2021). Rapid genomic testing in critically ill-children is also advancing the field of precision diagnostics and therapy in paediatrics. The first national healthcare system-funded implementation of rapid genomic sequencing for acutely unwell children commenced in England on 1st October 2019 led by the NHS Genomic Medicine Service. The diagnostic yield was 38% with the molecular diagnosis directing management in 94% of patients (Stark and Ellard, 2022).

Next-generation cytogenetics such as optical genome mapping (OGM) has also advanced the speed and ability of detecting genetic abnormalities (Lam et al., 2012). Cytogenetics is the genetic discipline that examines chromosomes for abnormalities. Karyotyping, chromosomal microarray analysis (CMA), multiple ligation-dependent probe amplification and fluorescent *in situ* hybridisation (FISH) are examples of currently utilised cytogenetic techniques but have limited resolution or limited region coverage. For example, CMA, despite having higher resolution for the detection of repeat regions of the genome known as CNVs, is limited in detecting balanced translocations and inversions (Levy and Wapner, 2018). Next-generation sequencing methods can detect sequence variants, but they are unable to accurately resolve CNVs. Moreover, complete genetic profiling in specific diseases either for diagnosis or for risk stratification of prognosis often relies on a combination of techniques which is time consuming and costly. OGM is a novel technology that analyses ultra-high molecular weight DNA molecules that provide a high-resolution genome-wide image. Each patient's unique map is aligned to a reference genome map to detect CNVs and structural anomalies. OGM has strong concordance with other cytogenetic techniques such as FISH and CMA but has also advanced the detection of additional chromosomal abnormalities that are clinically relevant and that are inaccessible to standard techniques, allowing for further disease stratification in diseases such as acute lymphoblastic leukaemia (ALL) (Lestringant et al., 2021; Rack et al., 2022). OGM is now recognised as key genomic technology capable of detecting all classes of structural variants in many disorders, which will undoubtedly improve the speed and accuracy of diagnosis of rare disorders in children (Levy and Wapner, 2018; Mantere et al., 2020) but also improve risk stratification in common haematological disorders in children such as ALL. Stratification in ALL has previously relied on a combination of genetic tests which is time consuming and expensive. In recent studies categorising subtypes of ALL, OGM increased the detection rate and cytogenetic resolution and abrogated the need for cascade testing using multiple cytogenetic techniques, resulting in reduced turnaround times and cost saving (Neveling et al., 2021; Rack et al., 2022). OGM has also been applied to other areas of paediatrics to improve the diagnostic yield including neurodevelopmental paediatrics (Shieh et al., 2020), myotonic dystrophy (Otero et al., 2021), disorders of sex development (Barseghyan et al., 2018) and with the potential for widespread application for prenatal diagnostic testing (Sahajpal et al., 2021). Given the extended capabilities of OGM, many undiagnosed conditions in paediatrics will be elucidated using this new technique and other

future developments as we enter an era of next-generation cytogenetics (Mantere et al., 2020).

## Developing biomarkers and targeted therapies to treat childhood diseases

### Proteomics

While advances in genomics and cytogenetics are revolutionising our understanding of diseases in childhood, these advances should not be considered in isolation. A systems biology or systems medicine approach is used to consider genetic mutations in the context of or interaction with other biological systems. *In vitro* and *in vivo* functional studies examine the translational effect of genetic mutations at a molecular level. Translational studies may include the large-scale impact on protein synthesis and function (proteomics) (Sahajpal et al., 2021) and metabolic function within the biological system (metabolomics). It is through the study of proteins within the sphere of proteomics that protein modifications can be characterised, and the targets of drugs identified. The value of proteomics within the context of precision diagnostics in children and adults is the understanding of the post-translational modifications that proteins undergo in response to a variety of intracellular and extracellular signals. Protein phosphorylation is important in protein signalling and disruption of protein phosphorylation due to alteration in protein kinases or phosphatases can lead to oncogenesis (Graves and Haystead, 2002). The process of cell growth, programmed cell death and the decision to proceed through the cell cycle are all regulated by signal transduction through protein complexes (Hunter, 1995). Dysregulation of these processes results in potential cancer development. Disruption of protein localisation can also have a profound effect on cellular function and can result in diseases in childhood such as cystic fibrosis (Graves and Haystead, 2002). Detailed knowledge about the structure, function, post-translational modifications, localisation, compartmentalisation and protein–protein interactions within cells supports the development of targeted drug therapies and the use of biomarkers to determine disease development and for monitoring treatment effect (Wilkins et al., 1996; García-Foncillas et al., 2021). However, proteomics provides an added layer of complexity as protein expression is altered by chronicity and environmental conditions (Al-Amrani et al., 2021). To enable drug development and the development of novel biomarkers, public resources of curated signal transduction pathways have been developed (Chatr-aryamontri et al., 2007; Mi et al., 2007; Schaefer et al., 2009; Kandasamy et al., 2010; Croft et al., 2011; Kanehisa et al., 2012; Kerrien et al., 2012; Franceschini et al., 2013; Holman et al., 2013; Schmidt et al., 2014). Proteomic-derived biomarker development is classified as diagnostic, predictive and prognostic, based on their uses.

Diagnostic biomarkers indicate if a patient has a specific disease, predictive biomarkers can predict the response to therapy, and prognostic biomarkers support the prediction of the clinical outcomes. Paediatric sepsis provides an example underpinning the value of using a precision diagnostic approach that sub-categorises patients to allow directed therapy to improve outcome and predict prognosis. Sepsis is a leading cause of mortality in children worldwide (Glaab et al., 2012). Given the heterogeneity of presentation in paediatric septic shock caused by viral or bacterial pathogens, this acts as an excellent model to demonstrate the value of proteomics in relation to precision diagnostic, predictive and prognostic biomarkers in children. Over a decade ago, whole blood RNA was first used to identify genes that were up- and down-regulated in a cohort of paediatric patients with septic shock, demonstrating that these patients had a unique gene expression signature (Wong et al., 2007; Liu et al., 2012). Early work identified three subclasses of children with septic shock who were defined by age, illness severity and complications classified by 100 genes; younger patients were more likely to have a higher level of illness severity, a higher degree of organ failure and a higher mortality rate (Cvijanovich et al., 2008; Wong et al., 2009). Further work in this field has led to the categorisation of children with septic shock into two endotypes which through iterations are now classified by only four genes (Wong et al., 2011, 2015). Of note, endotype A was noted to have repression of genes corresponding to glucocorticoid receptor signalling and thus adaptive immunity, and use of adjunctive corticosteroids in this group was associated with a 4-fold increase in mortality, thus helping to determine which children with septic shock would benefit from adjunctive corticosteroids and which would not (Wong et al., 2011, 2017). In addition, multiple serum protein biomarkers with known biological mechanisms in combination with mRNA biomarkers have been used in mortality risk stratification of children with septic shock (Wong et al., 2016). Combining metabolomic and inflammatory protein mediator profiling early after presentation of paediatric sepsis may differentiate children with sepsis requiring intensive care from those with or without sepsis who can be safely cared for without intensive intervention, thus enabling diagnostic and triage decisions in children with sepsis (Wong et al., 2017). Recent work has also helped to define the presence of bacterial or viral pathogens based upon the presence of biomarkers including procalcitonin, neutrophil gelatinase-associated lipocalin-2 and resistin (Mickiewicz et al., 2015) with further advances using host RNA signatures to further discriminate between bacterial and viral infection (Nijman et al., 2021). Other developments in this field include the use of proteomic profiling to distinguish late onset sepsis and necrotising enterocolitis in neonates (Pennisi et al., 2022), potentially predictive, early diagnostic and prognostic metabolomic and proteomic biomarkers in neonatal necrotising enterocolitis (Chatziioannou et al., 2018), and the combination of metabolomic, transcriptomic, genomic and proteomic profiles as an 'integrated-omics' systems medicine approach to the stratification, diagnosis, treatment and prognostics in paediatric sepsis (Agakidou et al., 2020). Outside the field of neonatal and paediatric sepsis, there are other examples where proteomics has the potential to further our understanding of conditions in children. In the field of paediatric allergy, novel proteomics methods have been used to identify the presence of 36 digested cow's milk proteins in breast milk which would be missed by immunochemical methods, and thus may help to improve our understanding of cow's milk protein allergy in exclusively breast-fed infants and to support new approaches to allergy prevention (Zhu et al., 2019). Recently, proteomics profiling as part of a multi-omics approach has been used to predict neurodegeneration in children with Down's syndrome pointing to a disruption of IGF1 signalling as a potential contributor to or biomarker to the neurodegenerative process, again potentially providing novel pathways for targeted treatments (Araya et al., 2022). In paediatric oncology, proteogenomic studies are furthering our understanding of relapse and treatment resistance in children with acute myeloid leukaemia, providing new approaches to predict relapse during disease progression, novel biomarkers and new targets for novel drug therapies (Stratmann et al., 2022). Other areas where proteomics provides value to prediction, diagnostics and personalised interventions include chronic kidney disease and peritoneal dialysis

in children (Cummins et al., 2022; Trincianti et al., 2022), drug metabolism and personalised dose optimisation in children (Streekstra et al., 2021; van Groen et al., 2022) and the use of multi-omics derived biomarkers to develop prediction models in children with asthma to guide therapy (Golebski et al., 2020; Kang et al., 2022). Proteomics either in isolation or part of a multi-omics approach has revolutionised our ability to develop individual disease profiling across multiple areas of paediatrics with new diagnostic, predictive and prognostic methodologies, and new targets for novel drug development.

### Precision diagnostics for targeted drug therapies

An understanding of genetic causes of specific diseases has resulted in the recent increase in the development of gene therapies in children regulated under the guidelines for advanced therapy medicinal products. Broadly, gene therapies were previously categorised into *ex vivo and in vivo* (Langley and Wong, 2017). *Ex vivo* is the route taken predominantly for the gene modification of bone marrow-derived cells and epidermal sheets. *Ex vivo* gene therapies are based upon the process of extracting immature bone marrow cells from the patient, employing a viral vector to integrate a functional copy of the gene into the genome of the target cells. The genetically modified cells are then re-administered to the patient in the form of an autologous gene-modified cell transplant. Examples of *ex vivo* gene therapies include the treatment of primary immune deficiencies and metabolic disorders (Qasim et al., 2007; Buckland and Bobby Gaspar, 2014) with more recent approvals granted for rare diseases such as metachromatic leukodystrophy (treated with atidarsagene autotemcel) (Rivat et al., 2012) and cerebral adrenoleukodystrophy (treated with elivaldogene autotemcel) (Fumagalli et al., 2022). The first UK patient to receive gene therapy in the UK celebrated his 21[st] birthday last year after having been treated for severe combined immunodeficiency (Keam, 2021). *In vivo* gene therapy primarily uses a viral vector (commonly the adeno-associated virus) to deliver the therapy directly to the patient rather than the transplant of gene-corrected cells. Glybera (alipogene tiparvovec) was the first gene therapy treatment to receive an European Commission marketing authorisation and the first gene therapy to be approved anywhere in the world, was used to treat lipoprotein lipase deficiency, an inherited condition with an incidence of 1/500,000. This therapy has since been withdrawn due to the rarity of the disease lack of cost-effectiveness (Great Ormond Street Hospital for Children (GOSH), 2021; Kastelein et al., 2013). There are currently five gene therapy treatments approved in Europe – Luxturna (for individuals with an inherited retinal disease caused by mutations in both copies of the *RPE65* gene), Zolgensma (to treat spinal muscular atrophy), two chimeric antigen receptor T cell therapies (Yescarta – to treat non-Hodgkin lymphoma and Kymriah for the treatment of adult patients with relapsed or refractory follicular lymphoma) and Strimvelis (the gamma-retrovirus for adenosine deaminase-severe combined immunodeficiency). Multiple other clinical trials are ongoing to identify candidate genes amenable to *in vivo* gene therapies (Ylä-Herttuala, 2012; Mendell et al., 2021). The recent advent of genome editing uses a different approach to correct genetic differences by introducing molecular tools to change existing DNA. Approaches include the use of zinc finger nucleases (Lee et al., 2021), transcription activator-like effector nucleases (Carroll, 2011) and clustered regularly interspaced short palindromic repeats (Liu et al., 2015), which has catalysed the implementation of new clinical trials of gene editing to cure rare and common diseases in

children that otherwise result in death in childhood or early adulthood (Lee et al., 2018, 2021).

Biomarker-driven directed therapies in the treatment of cancer in children are still early in development. A recent systematic review reporting on the clinical utility of precision medicine in the treatment of paediatric cancer has reported on the use of molecular techniques such as array comparative hybridisation, immunohistochemistry, next-generation sequencing, RNA sequencing, single nucleotide polymorphism array, targeted panel-based sequencing, whole-exome sequencing and WGS to identify genomic targets, which have guided the allocation of targeted drugs (Salsman and Dellaire, 2017). Genes which are transcribed in any one condition are known as the transcriptome; the process of determining the genetic codes contained in the transcriptome and their relative proportions is known as transcriptome sequencing or transcriptomics. Of note, the value of understanding the transcriptome has been highlighted through the increased benefit of identifying additional therapeutic targets that were not identified by genomic analysis alone (Marks et al., 2017; Uddin et al., 2020). Targetable mutations were found in 48.0% of patients, with 41.7% who received targeted drugs demonstrating an objective response. However, accessibility and thus inequity of access to targeted oncological therapies were cited as problematic, with only 27% of patients receiving targeted treatments, highlighting the gulf between access to precision diagnostics and subsequent therapies (Salsman and Dellaire, 2017). Challenges cited included the need to access drugs off-licence due to lack of paediatric dosing schedules, lack of access to clinical trials at their treating centre or ineligibility owing to advanced disease or trial restriction to adult patients or the dependency on compassionate access facilitated by pharmaceutical companies (Salsman and Dellaire, 2017). Moreover, heterogeneity in outcome reporting across clinical trials may lead to bias in the interpretation of treatment effect. This underpins the need for consistency in clinical trial outcomes to clearly define the clinical and economic value of a precision diagnostics approach to targeted therapy. Large molecular databases and global collaborative clinical trials have been assembled to help demonstrate the value of targeted therapies which will in turn enhance our understanding in the future and improve access to novel therapies (Pincez et al., 2000; Brien et al., 2016; Harttrampf et al., 2017; Linzey et al., 2018; George et al., 2019; Hansford, 2019; Gojo et al., 2020).

### A digital approach to precision diagnostics and medicine in paediatrics

### The value of population data

The focus on precision diagnostics in paediatrics has largely focused on the value of data derived from 'omics' profiling as a means of supporting diagnoses, risk stratification, directing therapy (pharmacogenomics and pharmacometabolomics) and determining prognostic indicators. While this approach is revolutionising the way patients are managed, a systems medicine approach should also consider data relating to patient populations and external factors that may impact on patient diagnosis, monitoring and care. While precision in the initial diagnosis enhances a personalised approach to therapy, longitudinal patient monitoring facilitates a dynamic precision approach to 'diagnosing' new or recurrent issues. As medical device development in paediatrics and child health gathers momentum (Vo et al., 2020), the value of personalised and population data to direct patient management is attracting attention. For example, large data sets relating to growth in children

combined with automated systems for monitoring linear growth are facilitating the early identification of growth disorders of children at a greater frequency while reducing the number of referrals of children with normal growth parameters (Dimitri, 2019). For patients diagnosed with disorders of growth requiring growth hormone therapy, large-scale data acquisition using a growth hormone delivery device with a connected monitoring platform (easypod™), which automatically transmits adherence data via an online portal (easypod™ connect), can be used to predict an individual's response to therapy relative to a population undergoing the same treatment, by integrating methodologies such as machine learning. This facilitates clinical decision support to modify therapy in individual patients (Sankilampi et al., 2013). This approach of using patient population data to 'diagnose' poor adherence to therapy has been a catalyst to develop a digital ecosystem to support service users and service providers in the management of growth disorders requiring growth hormone therapy. This includes the development of online educational materials to support digital literacy and patient management (Dimitri et al., 2000; Su et al., 2022), digital monitoring of patient therapy, adherence and preferences (Boman et al., 2021; Spataru et al., 2021; Spataru et al., 2022; van Dommelen et al., 2018; Koledova et al., 2020) and the development of a framework to guide future digital developments including automated referral pathways, enhanced digital communication, digital medical and psychological support, gamification to support adherence to therapy, access to digital resources, digital reporting of patient reported outcomes and safety and assessment reporting (Dimitri et al., 2021).

### The application of machine learning in paediatric diagnosis

The advent of machine learning has facilitated the use of large patient data sets or 'training sets' to generate and refine predictive models. Large data sets already exist in multiple areas of medicine, and these are now being employed to predict disease in children. Machine learning has superior capabilities to traditional data analytics methods based upon the ability to rapidly process large volumes of complex data, to explore and extrapolate data relationships through pattern recognition that are not recognisable through other methods and improving efficiency and accuracy through data acquisition. Deep learning lets the data train the computer leading to predictive models that become stronger as more data are added. Imaging data sets that are being used in paediatrics to predict abnormalities have been frequently reported in the literature in the last few years with examples including (but not limited to) the detection of abnormalities in chest radiographs (Chen et al., 2020; Padash et al., 2022), the diagnosis of effusions in elbow joints (Huhtanen et al., 2022), the detection of intracranial pathology on CT imaging, defining abnormalities as critical or non-critical (Titano et al., 2018), the assessment of left ventricular function from birth to 18 years and the diagnosis of coronary artery lesions in Kawasaki disease using echocardiography (Lee et al., 2022; Zuercher et al., 2022) and the assessment of paediatric brain tumours (Grist et al., 2021; Huang et al., 2022). However, challenges remain in achieving an acceptable diagnostic accuracy and the acquisition of adequate training data sets in cohorts, particularly in relation to rare diseases due to the size of the patient populations. Machine learning has been applied to continuous conventional electroencephalography to determine the diagnostic accuracy of detecting and monitoring seizures in neonates. The artificial intelligence (AI) platform did not improve the number of neonates diagnosed with seizures, although the quantification of seizure burden was greater, thus providing a more accurate means of monitoring seizure frequency and duration (Pavel et al., 2020). The DeepGestalt platform utilising computer vision and deep-learning algorithms for facial image analysis has shown great potential in the phenotypic evaluation of syndromes including the initial syndromic diagnosis and the ability to subclassify different genetic subtypes within the same syndromic diagnosis (Gurovich et al., 2019). Patient population data have also been used to detect the earlier presentation of more common presentations in paediatrics such as neonatal sepsis (Masino et al., 2019), the risk stratification of infants with bronchiolitis to promptly identify infants at risk of deterioration (Raita et al., 2020), improving the prediction of clinical outcomes in children presenting to the emergency department with the benefit of better identifying critically ill children while reducing the over triaging of children who are less ill (Raita et al., 2020) and predicting the need for hospitalisation of children with asthma (Patel et al., 2018). Notably, this last example also demonstrates the value of using environmental data including weather data, population influenza patterns and socioeconomic status to improve predictability. The value of data acquisition from social media platforms on smartphones and tables to support the diagnosis of mental health issues in young people and adults such as depression (Reece and Danforth, 2017), post-traumatic stress disorder (Reece et al., 2017) and schizophrenia (Hänsel et al., 2021) highlights the value of 'non-health' data to support diagnoses. Given that children and young people are high users of technology and social media, it is important to consider the value of personalised data on commercially available platforms in the future to inform on health, well-being and disease. For example, Instagram posts have been used as a predictor of physical activity in adults aged 18–30 years (Liu et al., 2021). A comprehensive review of machine learning in paediatrics is covered elsewhere (Clarke et al., 2022), supporting the value of AI in the field of precision diagnostics in paediatrics. As electronic patient records become the global norm, it is likely that the application AI will become fundamental to improve diagnosis and clinical outcomes. However, large and accurate data sets will be required to ensure accurate outputs from machine learning models, and legal and governance structures will need to be in place to regulate the use of these data. Given the relative paucity of large paediatric data sets, interoperability across systems and continental or global data sharing of high-quality data may be required to optimise outputs but will also need to be viewed in the context of generalisability across different healthcare systems.

### Future perspectives

Advances in science, technology and data acquisition and analytics have already revolutionised paediatric healthcare. Adopting the '4P' systems medicine approach (Hood et al., 2004; Flores et al., 2013) to ensure healthcare is preventive, predictive, participatory and personalised will rely on the combination of multiple data sources and methodologies to maintain health and prevent and predict disease. The value of combining 'omics' data has advanced the science of molecular precision medicine to understand the molecular basis of disease, the risk of developing it and its evolution and pathogenesis in children; the speed of technological advances in genomics, proteomics, transcriptomics and AI will further improve this (Williams et al., 2018; García-Foncillas et al., 2021).

The future of precision diagnostics in paediatrics will in part rely on data acquisition that has not been commonly used in medicine before from commercial and social media platforms, and from

environmental sources, knowing that the lifestyle, demographic and environmental factors play a key role in the presentation and course of disease. Personalised platforms on mobile phone and tablets, digital monitoring systems within homes and remote sensing technologies can gather data on personal habits, lifestyle, physical activity and diet, and environmental data such as climate data can be derived from institutional sources. Collectively, these factors are encapsulated in the concept of the 'exposome'. The 'exposome' has been proposed as a new paradigm to encompass the totality of human environmental (non-genetic) exposures from conception onwards (Wild, 2012; Vrijheid, 2014). Thus, the exposome will play a fundamental part in precision diagnostics through the life course. In this context, there is a value in collecting data from early life to truly reflect response to changes in the exposome which in turn support disease prevention, predicting the future risk of developing a disease and supporting the management of disease in an environmental context once established. To facilitate this, advances in sensing devices and other technologies will facilitate collection and remote access to these data (Martin Sanchez et al., 2014). In addition to the challenges of collecting and extrapolating data, data analytics and combining data sets, the context in which these data are derived must be considered. Despite the inherent affinity children and young people have with technology, there is a reluctance in this population to simply 'hand over' data, and data acquisition must be contextualised to social context. Children and young people are reluctant to use technology to collect health data if this does not fit in with social norms, thus risking discrimination. Concerns have been raised surrounding data sharing, use and confidentiality of personal information, with children and young people needing reassurance that their data are being used by trusted organisations, that it is stored safely and securely, with a desire to control their data and privacy, and to minimise the risk of misinterpretation of their health data, particularly in the context of them having a health-related problem (Blower et al., 2020). Participation is framed as a central tenet of personalised medicine. As we move in the direction of whole-scale personalised medicine that involves the collection and storage of data in children, particularly in relation to the future predictability or certainty of disease, the ethical and legal considerations around informed consent, data protection, autonomy and privacy must be considered carefully, particularly for those who are not old enough to express their views. There are potential legal and ethical ramifications from extensive testing and use of data that may have unintended consequences, for example, the identification of a secondary disease that does not manifest until adulthood, that in turn may limit testing in children. Thus, guidelines on how to pursue children's participation in personalised medicine would be of benefit in the future (Ó Cathaoir, 2021).

## Conclusion

The way data is changing health and healthcare in the 21$^{st}$ century could be likened to the impact that the introduction of antibiotics had on bacterial disease in the early 20$^{th}$ century – it will save and improve many lives. We have already seen significant advances in diagnosis, drug discovery and interventions in healthcare as a result of advances in the 'omics' and cytogenetics. The future acquisition and combination of data from multiple sources including data from individuals, their lifestyle behaviours and their immediate and wider environments will add to our understanding of the manifestation of health and disease over time. A life course approach starting from birth or even in the prenatal period is required to understand how human development and changes in the environment lead to future health and ill-health, and how environmental and physical factors impact on congenital and acquired diseases of childhood. As we advance in the field of personalised medicine, we need to consider a paradigm shift in precision diagnostics to incorporate the need to 'sustain health' in addition to accurately diagnosing disease. In this respect, this shift in diagnostics should encompass the need to 'diagnose' good health. The principle of 'start early and change more' underpins the value of applying personalised medicine early in life for a better future.

**Open peer review.** To view the open peer review materials for this article, please visit http://doi.org/10.1017/pcm.2023.4.

**Competing interest.** The author declares no conflict of interest.

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
