## [Reviewer Report]

*Comments to Author*: 1- English: The manuscript could benefit from editing for grammar, missing words, and subject-verb agreement, etc. It is recommended that authors delete irrelevant "general" phrases and sentences, repeated and unneeded words. They should use short sentences. Also, some Introductory sentences are irrelevant or are not needed. There are also typos in the manuscript.

2- Abbreviations: All abbreviations should be revised and defined at their first use.

3- Abstract: The abstract is very general and does not tackle pediatric population or diseases at all. The main purpose of the review is not clearly defined in the abstract. What is actually mentioned are all known facts that precision medicine and patient centered care is the future.

4- Introduction: “participatory component of P4 medicine” it is confusing what is meant by P4? Is it the same as 4Ps?

5- Introduction: “data capture from an early age will rely on empowering children and young people to contribute their data, motivating them to maintain this activity and individual trust that data will be utilised appropriately and stored secure.” What exactly do those data encompass? Please elaborate and explain how individuals can contribute their data?

6- Introduction: talking about sharing personal health data from early childhood to adulthood, isn’t this breach of patient privacy? What protects this privacy?

7- Introduction: there is no mention of a single word related to pediatrics in the introduction. The author should have a good flow of ideas where he jumps from one section to another using liaisons.

8- Section “Disease reclassification and predicting future disease in children”: in this section, I do not feel the authors provided any interesting finding that could be beneficial for readers. It is well known that reclassification of many diseases and cancers based on molecular signatures is a reality we are living, such as brain tumors.

9- Under “Developing biomarkers and targeted therapies to treat childhood diseases”: the author also tackled proteomics, predictive biomarkers, and precision diagnostics for targeted drug therapies, from a very broad perspective. In the section proteomics, the author just defined what proteomic medicine is about, without giving examples from the pediatric world. Similarly, under predictive biomarkers, not a single example is provided.

---

## [Editor Report]

*Comments to Author*: I agree that the details that Reviewer #1 points out all do have some truth to them, however I think they are in general a little overstated in their severity and hence I believe the article can be ready to publish with only some revisions satisfying the spirit behind Reviewer #1 comments and my additional review points below. Overall, the article is a highly relevant and beneficial contribution to an important aspect of precision medicine, that of how it impacts the health of patients early in life.

• Example: yes, the author interchanges P4 and 4P, most often using what I believe the non-standard abbreviation (and as used in the cited references). However, they do define their abbreviation at the first use. Simple to fix. Otherwise, I don’t see an abnormal rate of grammatical errors.

o Page 4: single-quotes mis-spaced in “individual data profile”

o Page 5: “revolutionise our ability to modify”

o Page 9: no hyphen for in vitro or in vivo (ex vivo on p11, “in turn” on 16)

• I agree that the author will make this good article even better by describing some additional examples as referred to by Reviewer #1 (especially the proteomics section, since concrete examples are provided in the subsequent section), but to be fair, this is a rather broad-reaching review and I think it is thoroughly blanketed in appropriate citations even if not fully discussed.

• I believe the article would do well to discuss in a bit further detail the inherent “problem” of germline genetic disease, where there may not be a cure/treatment…and the primary benefit is either to prepare families for appropriate palliative care or to rule out other more actionable diseases. This is a common conundrum/issue in the field, so addressing it head on in more depth than currently mentioned in the reclassification section would be good.

• I’m slightly concerned about the absence of specific/depth of coverage of next generation cytogenetics methods (eg., CMA or optical mapping), but perhaps the author has good reason to omit this contribution to the completeness of this review (as I recognize the breadth of the review inherently limits how much depth can be covered).

• I feel the author’s pointing out the need for “legal and governance structures” is sufficient counterbalance to address Reviewer #1’s concerns in this area. Perhaps the concern hints at a benefit of pointing out that the solutions will likely vary based on localised laws and culture.

---

## [Editor Report]

*Comments to Author*: This manuscript, and especially the additional examples included in this revision, is a very nice review of the state of and important benefits of precision medicine in the context of childhood disease.